# Mean Temperature and Drought Projections in Central Africa: A Population-Based Study of Food Insecurity, Childhood Malnutrition and Mortality, and Infectious Disease

**DOI:** 10.3390/ijerph20032697

**Published:** 2023-02-02

**Authors:** Munum Hassan, Kinza Saif, Muhammad Saad Ijaz, Zouina Sarfraz, Azza Sarfraz, Karla Robles-Velasco, Ivan Cherrez-Ojeda

**Affiliations:** 1Department of Research, Akhtar Saeed Medical College, University of Health Sciences, Lahore 54000, Pakistan; 2Department of Research, Wah Medical College, University of Health Sciences, Wah Cantt 47000, Pakistan; 3Department of Research, Rawalpindi Medical University, Rawalpindi 46000, Pakistan; 4Department of Research and Publications, Fatima Jinnah Medical University, Lahore 54000, Pakistan; 5Department of Pediatrics and Child Health, The Aga Khan University, Karachi 74800, Pakistan; 6Department of Allergy and Pulmonology, Universidad Espíritu Santo, Samborondón 092301, Ecuador

**Keywords:** climate change, drought, mean temperature, Central Africa, food insecurity, malnutrition, mortality, infectious disease, global health

## Abstract

The Central African Region is an agricultural and fishing-based economy, with 40% of the population living in rural communities. The negative impacts of climate change have caused economic/health-related adverse impacts and food insecurity. This original article aims to research four key themes: (i) acute food insecurity (AFI); (ii) childhood malnutrition and mortality; (iii) infectious disease burden; and (iv) drought and mean temperature projections throughout the twenty-first century. Food insecurity was mapped in Central Africa based on the Integrated Food Security Phase Classification (IPC) for AFI. The global hunger index (GHI) was presented along with the proportion of children with undernourishment, stunting, wasting, and mortality. Data for infectious disease burden was computed by assessing the adjusted rate of change (AROC) of mortality due to diarrhea among children and the burden of death rates due to pneumonia across all age groups. Finally, the mean drought index was computed through the year 2100. This population-based study identifies high levels of hunger across a majority of the countries, with the mean drought index suggesting extreme ends of wet and dry days and an overall rise of 1–3 °C. This study is a source of evidence for stakeholders, policymakers, and the population residing in Central Africa.

## 1. Introduction

Central or Middle Africa is a subregion of the African continent that comprises various countries; these include Angola, Cameroon, the Central African Republic, Chad, the Democratic Republic of the Congo, the Republic of the Congo, Equatorial Guinea, Gabon, and Sao Tome and Principe [1,2,3]. A large proportion of the Central African population’s economy thrives on agricultural and fishing activities; an estimated 40% of the population resides in a rural setting and lives in poverty [4,5,6]. A common development challenge in the entire African region is the burden of disease and the health status of residents [7]. While the average life expectancy of an individual residing in the Southeast Asian region is 68 years, the life expectancy in Africa is 58 years, as per the World Health Organization [8,9]. The life-expectancy gap has increased since the 1980s, particularly due to diseases including the human immunodeficiency virus (HIV), tuberculosis, Ebola, malaria, SARS, and SARS-CoV-2. The fundamental concerns are low economic performance, poor pediatric health, and the overall burden of morbidity and mortality in Central Africa [10,11,12].

The State of the Climate in Africa 2019, a report by the World Meteorological Organization (WMO), reports current climate trends and possible impacts on agriculture and the economy [8]. Increasing threats to water and food security, human health, and social-economic development in Africa are anticipated by 2050 [13,14,15,16]. 2019 was one of the warmest years for the continent; the mean temperature in Africa has been warming at a comparable rate to other continents; however, it has been warming quicker than the global mean surface temperature [17,18]. As compared with 1901, most of Africa has already warmed by more than 1 °C; an increased frequency of hot days and heatwaves is imminent [17,19,20]. In African countries that are drought-prone, the Food and Agricultural Organization (FAO) finds that undernourishment has increased by 45.6% [8]. If no safeguarding measures are taken, the mean yield of agricultural practices will reduce by 13% in Central and West Africa, whereas the trends are 11% in North Africa and 8% in South and East Africa [8]. As one of the most impacted subregions in Africa, any median increases in temperature along with changes in rainfall patterns will increase vector-borne diseases including yellow fever, malaria, and dengue fever [19,20,21].

The International Monetary Fund (IMF) reports that the negative impacts of climate change are concentrated in regions with hot climates, particularly regions with low-income residents [22,23,24]. The African Climate Policy Center also predicts that the gross domestic product in all African regions will largely decrease with the ongoing rise in global temperature [8,25]. With scenarios of a 1–4 °C rise in temperature, the GDP can fall by 2.25–12.12% [8]. Africa’s Agenda 2063 recognizes that climate change has been a central challenge in the continent’s development [26,27]. Since 2015, the Nationally Determined Contributions (NDCs) aligned with the Paris Agreement have been somewhat instrumental in guiding policies against climate change. A total of 52 countries in Africa submitted NDCs [28,29,30].

Central Africa may be viewed distinctly from the rest of the globe, even from other low- and middle-income countries. Dangerous agents oftentimes emerge or re-emerge in the African context. Particularly, there are difficulties with morbidity and mortality surveillance in Central Africa. Concerning the various epidemics in the region that have become more evident in the past decade, the surveillance systems have been established in localities have revealed multifactorial issues. These include the following: microbial change and adaptation; humans who are susceptible to infection; changing ecosystems, climate change, and weather extremes; human behavior and demographics; land use and economic development; social unrest and poverty; famine; and political instability [31]. The majority of these factors are overrepresented in the Central African region. Therefore, many emerging diseases, one of which is HIV, and another, malaria, have emerged from wild monkeys in the region. Many bacteria, including *Rickettsia felis*, *Tropheryma whipplei*, *Yersinia pestis (plague)*, *and Vibrio cholera (cholera)*, have emerged in Africa in the 21st century [32]. Similarly, the following viruses have emerged and reemerged in the region: measles, yellow fever virus, Ebola, monkeypox (i.e., a global threat), rift valley fever, chikungunya virus, and zika virus [32].

The motivation behind this study is to establish, synthesize, and present collated findings connecting the outcomes of drought in a grossly under-neglected region—Central Africa. Population-based climate change studies have been published in the literature for regions including South Asia and East Africa [33,34,35,36]. Population-based studies are defined as a group of individuals from the general population who share common characteristics, such as age, sex, or health conditions [37]. However, with the paucity of studies in Central Africa and the large brunt the region has faced and will face in the future, this study ties in climate change outcomes with the prevalence of food insecurity, childhood malnutrition, and infectious disease connections in the populace. This original research article aims to connect four themes (i.e., food insecurity, malnutrition, infectious disease, and climate change) and provide a past, present, and future population-based analysis of climate impacts, health, and food trends in the Central African region.

## 2. Materials and Methods

Four central themes were quantified in this paper. They included (i) food insecurity, (ii) malnutrition indicators and trends, (iii) infectious disease indicators and trends, and (iv) mean air temperature and drought projections in the Central African Region. These were connected with the framework proposed by the World Health Organization [38], which linked the climate crisis to the health crisis. It is depicted in Figure 1.

The cumulative data spanned 1990 through 2022, and projections were made until 2100. However, specific outcome data had different origins and end points, which are listed separately in the results section for every variable of interest. The databases utilized to collect population-based data are listed in Table 1.

Countries in Central Africa were coded by the Integrated Food Security Phase Classification (IPC) for both acute food insecurity (AFI) and chronic food insecurity (CFI) scales. The analysis used was GeoJSON, which is an open standard format designed for geographical feature representation and non-spatial attributes. Data were obtained from the IPC [39]. The findings of acute food insecurity were presented in an IPC-coded geographical map of the region.

The global hunger index (GHI) was the composite tool used that measured and tracked hunger at the national, regional, and global levels; it reflected multiple dimensions of hunger over time. The datasets from Concern Worldwide and Welthungerhilfe [40] were utilized to synthesize results and present GHI findings. The calculation was made on a 100-point scale, with 0 indicating the best score and 100 being the worst score. There are four indicators that were combined to capture the multidimensional origin of hunger. These four indicators, as listed below, are used to measure the progress of hunger mitigation efforts toward the United Nations Sustainable Development Goals (SDGs). These comprised the following:1.**Undernourishment:** The proportion of the population with inadequate calorie intake.

Prevalence of undernourishment/80 × 100 = standardized undernourishment value

2.**Child stunting:** The proportion of under-five children with a low height for age (HAZ), indicating chronic undernutrition.

Child stunting rate/70 × 100 = standardized child stunting value

3.**Child wasting:** The proportion of under-five children with low weight for height (W/H), indicating acute undernutrition.

Child wasting rate/30 × 100 = standardized child wasting value

4.**Child mortality:** The percentage of children who die before the age of 5, indicating unhealthy environments and inadequate nutrition.

Child mortality rate/35 × 100 = standardized child mortality value

Data for infectious disease burden of pneumonia and diarrhea. For pneumonia, a country-by-country breakdown was given for pneumonia mortality rates in Central Africa between 1990 and 2019 was given, presented as the annual number of deaths from pneumonia per 100,000 people in different age groups. Datasets by the Institute for Health Metrics and Evaluation, Global Burden of Disease (2019), were utilized [41]. For diarrhea, the estimates of prevalence, incidence, and diarrhea-related mortality of under-five children across countries in Central Africa between 2000 and 2015 were utilized. The estimates were produced with data obtained from censuses and many household survey series, including the Multiple Cluster Survey (MICS), the Demographic and Health Survey (DHS), and other country-specific surveys. This comprised GeoTIFF raster files that provided several indicators of diarrhea, which were coded to generate estimates. The primary data was obtained from the Institute for Health Metrics and Evaluation (IHME) [42]. The adjusted rate of change (AROC) of mortality due to diarrhea was thereby plotted under the findings. In general, the AROC represents the momentum of the variable, where a positive value reflects increased mortality due to diarrhea and a larger negative value reflects the opposite change in direction of mortality.

The World Bank’s climate change knowledge portal was used for country-specific drought outcomes [43]. The analysis was tailed by selecting projected socioeconomic pathways (SSPs) that provide insight into future climates based on defined missions, mitigation efforts, and development pathways. SSPs are a component of the new framework adapted in climate change research to facilitate integrated analysis of future climate impacts, adaptation, vulnerabilities, and mitigation. It must be stated that SSPs are scenarios of projected socioeconomic global changes up to 2100. Further clarification for the SSP scenarios is depicted in Figure 2.

In accordance with the IPCC 6th Assessment Report (AR6), the projected air temperature outcomes of the five scenarios are based on the SSPs (Figure 1). The names of the scenarios are then combined with the expected radiative forcing in the years until 2100 (1.9 to 8.5 W/m^2^) [44]. This leads to the computation of the scenario names (i.e., SSP_x-y.z_) as depicted in Table 2. 

The database accounts for the following: (i) population by sex, age, and education; (ii) urbanization; and (iii) economic development. The forecasting until the year 2100 is not only limited to the basic SSP socio-economic elements but also includes scenarios from integrated assessment models (IAMs). These scenarios also account for (i) land use, (ii) energy supply and use, (iii) greenhouse gas emissions and air pollutant emissions, (iv) average air temperature change, and (v) mitigation costs. The climate projection is based on modeled data from the global climate compilations of the coupled model intercomparison projects (CMIPs). The projection data is presented at a resolution of 1.0° × 1.0° (100 km × 100 km).

The SPI is calculated using monthly (or weekly) precipitation as the input data. The SPEI uses the monthly (or weekly) difference between precipitation and PET. This represents a simple climatic water balance that is calculated at different time scales to obtain the SPEI. The standardized precipitation evapotranspiration (SPEI), also called the mean drought index, and the mean air temperature values were calculated through the year 2100 and individually reported for the years 2030, 2070, and 2100 using the low-end range (SSP1–1.9) and the high-end range (SSP5–8.5) [43]. The values were reported with interquartile ranges (IQR), wherever applicable. The IQR is a measure of statistical dispersion, which was utilized to report the spread of the data findings. The input variables, application, and key resources of the SPEI are presented in Table 3 [45].

The moisture category is interpreted as follows: *2 and above* (extremely wet, EW); *1.5 – >1.99* (very wet, VW); *1 – >1.49* (moderately wet, MW); −0.99 – >0.99 (near normal, NN); *−1 − > −1.49* (moderately dry, MD); *−1.5 – >−1.99* (severely dry); and *−2 and less* (extremely dry, ED). The SPEI is closely related to crop and water resources and ecosystems. Thereby, the SPEI accounted for potential evapotranspiration and precipitation in determining drought outcomes in the Central African region.

## 3. Results

### 3.1. Food Insecurity in the Central African Region

Around 2.7 million (44%) people in the Central African Republic are experiencing and are expected to experience high levels of acute food insecurity, which is labeled a crises and emergency (IPC Phases 3 and 4) between September 2022 and March 2023. These trends are driven by some dry spells and severe flooding, which have led to acute food insecurity. In total, over 2 million people are in IPC Phase 3 of the crisis, whereas 0.642 million are in IPC Phase 4 of the emergency. The highest food insecurity is typically classified as IPC Phase 3 or above, and the projections are made primarily for those living in poor households and peri-urban or rural areas with low food purchasing power (Figure 3).

The acute food insecurity situation between July and December 2022 in the Democratic Republic of the Congo (DRC) impacted 26.4 million residents where high levels of acute food insecurity (IPC Phase 3 or above) were witnessed (Figure 1). The analysis indicates that out of 26.4 million residents, 22.6 million are in IPC Phase 3 crisis, whereas 3.8 million residents are in IPC Phase 4 high food insecurity—these trends represent 21% of the population living in the DRC. The country has the largest proportion of food insecurity in the world, which has resulted from a myriad of factors, including increased food prices, conflict, high transportation costs, the COVID-19 pandemic, and other epidemics in the region. Projections until June 2023 indicate that there may be a slight decrease in food insecurity, with 24.5 million people across 107 localities expected to reach IPC Phase 3 crisis; however, areas such as Ituri and North Kivu may experience IPC Phase 4 emergency given the country’s ongoing conflicts. The national level of food security is worsened by increased fuel prices and pre-existing poor infrastructure.

Cameroon’s food and security situation between March and May 2022 comprised 9% (2,413,288) of the population in the IPC Phase 3 crisis. Similarly, Chad’s situation between March and May 2022 comprised 14% (2,098,861) of the populace in an IPC Phase 3 acute food crisis (Figure 3). Data could not be obtained due to inadequate evidence from other countries in the region.

Chronic food insecurity data was unavailable for all countries in the Central African Region, excluding the DRC, for the years 2016–2021. On average, 65% of adults in the country eat two meals a day, whereas in Tshopo and Western Kasai, 51% and 71% of residents, respectively, eat one meal a day. The vitamin A supplementation among the pediatric population was overall satisfactory, excluding Tshopo (44%), Sankuru (41%), Maniema (40%), Nord Ubangi (47%), and Sankuru (41%), where low coverage rates were present. However, drinking water distribution has been close to non-operational with zero access in certain areas; only the Kinshasa province witnessed a rise in households with access to managed water from 87% in 2013 to 37% in 2013.

### 3.2. Malnutrition Indicators and Trends in the Central African Region

#### 3.2.1. Global Hunger Index (GHI)

Of the six countries that were analyzed, two had alarming levels of hunger, two had serious hunger levels, and two had moderate levels of hunger (Figure 4). While Angola had a GHI score of 64.9 in 2000, the value reduced to 25.9 in 2022, which still indicated that there was serious hunger; the country ranks 98th out of 121 countries. Cameroon had 35.8 GHI scores in 2000, and the value was 18.9 in 2022, which indicates that the hunger level is moderate; Cameroon ranks 80th out of 121 countries. With a 48.8 GHI score in 2000 in the Central African Republic, the country had no large shifts by 2022 with a score of 44; the level of hunger is alarmingly high, and the country ranks 120th out of 121. Chad had a GHI score of 50.7 in 2000, and the score reduced to 37.2 in 2022, meaning that the level of hunger was alarming; the country ranks 117th out of 121. The Republic of the Congo had a GHI score of 34.7 in 2000, which was reduced to 28.1 in 2022, indicating serious hunger issues and a country rank of 105 out of 121. Gabon had a GHI score of 20.9 in 2000, and the score reduced to 17.2 in 2022, indicating moderate hunger levels; the country ranked 76th out of 121. No data were available for Equatorial Guinea and São Tomé and Principe.

#### 3.2.2. Undernourishment, Wasting and Stunting, and Under-Five Mortality

Three countries witnessed a decline in undernourishment, whereas the other three reported a rise in the proportions of the population (Figure 5). Angola’s undernourished population proportion decreased by nearly three times, from 67.5 to 20.8 between 2000 and 2022. Cameroon also witnessed a similar decline in undernourishment from 22.9 to 6.7 in the same period. The Central African Republic witnessed a rise in undernourishment from 39.2 in 2000 to 52.2 in 2022. Chad witnessed a modest decline from 38.8 to 32.7 in the same period. The Republic of the Congo also saw a rise, from 27 in 2000 to 31.6 in 2022. Gabon’s undernourishment rate increased from 10.7 to 17.2 in the same period. No data were available for Equatorial Guinea and São Tomé and Principe.

All countries reported a reduction in the prevalence of wasting in children under five years (Figure 5). Angola’s wasting prevalence for children under five decreased from 11.2 to 6.1 between 2000 and 2022. Cameroon also saw a decline from 6.2 to 4.3 in the same period. The trend was followed by the Central African Republic, with a nearly 50% reduction in the prevalence of wasting from 10.4 to 5.3. Chad’s trends fell from 13.9 to 10.2, which was similar to the Republic of the Congo reporting values of 9.8 in 2000 and 7.9 in 2022. Finally, Gabon also witnessed reductions in wasting among children, from 4.2 in 2000 to 3.3 in 2022. No data were available for Equatorial Guinea and São Tomé and Principe.

All countries had a reduction in the prevalence of stunting in children under five years (Figure 5). Angola saw a fall to 29.8 in 2022 from 46.7 in 2000. Cameroon also had similar reductions, from 38.2 to 28.9, in the same period. The Central African Republic had negligible reductions from 44.4 to 40 between 2000 and 2022. Chad’s prevalence of stunting fell from 38.9 to 31.1 in the same period. The Republic of the Congo had an overall comparable prevalence in 2000 (29.9) and 2022 (26.4). Finally, Gabon witnessed a modest decline in stunting prevalence (from 25.9 to 17.8 from 2000 to 2022). No data were available for Equatorial Guinea and São Tomé and Principe.

Notably, the under-five mortality rate fell in all six countries (Figure 5). In Angola, there was a three-fold reduction from 20.4 to 7.1 in 2000 and 2022. Cameroon saw a nearly half-fold fall in mortality rates from 14.4 to 7.1 in the same period. The Central African Republic’s mortality rate was 10.3 in 2022, which fell from 16.9 in 2000. The highest current under-five mortality rate was witnessed in Chad, with 11 values in 2022, which fell from 18.4 in 2000. The Republic of the Congo had a 4.5 mortality rate in 2022, which fell from 11.4 in 2000. Gabon’s mortality rate also fell by half, from 8.3 in 2000 to 4.2 in 2022. No data were available for Equatorial Guinea and São Tomé and Principe.

### 3.3. Indicators of Infectious Disease

#### 3.3.1. AROC of Mortality Due to Diarrhea between 2000 and 2015

The adjusted rate of change (AROC) of mortality due to diarrhea between the years 2000 and 2015 was calculated for Angola, Cameroon, Chad, Congo, the Democratic Republic of the Congo, Equatorial Guinea, and Gabon (Figure 6). Overall, the mean AROC of mortality due to diarrhea was the largest in Angola (AROC = −0.134), which was followed by Equatorial Guinea (AROC = −0.104), and Cameroon (AROC = −0.084). Whereas, the slowest rate of change in mortality due to diarrhea was witnessed in the Central African Republic (AROC = −0.008), which was followed by Gabon (AROC = −0.025), Chad (AROC = −0.046), Congo (AROC = −0.048), and the Democratic Republic of the Congo (AROC = −0.056).

#### 3.3.2. Pneumonia Mortality Rates by Age between 1990 and 2019

The mortality rates due to pneumonia were overall low among the 5–14 and 15–49 age groups. Of note were the downward trends in under-5-year mortality trends in the year 2019 in Angola, the Democratic Republic of the Congo (103 per 100,000), Equatorial Guinea (39.6 per 100,000), and Sao Tome and Principe (62.2 per 100,000). The highest under-5-year mortality due to pneumonia in 2019 was noticed in Chad (412.6 per 100,000), which was followed by the Central African Republic (388.6 per 100,000) and Cameroon (185.2 per 100,000). A pictorial representation of pneumonia mortality rates by age across all countries in Central Africa between 1990 and 2019 is given in Figure 7.

### 3.4. Statistical Trends of Air Temperature Changes and Drought Outcomes through 2100

#### 3.4.1. Statistical Reporting of Temperature Change Predictions between 2030 and 2100

For Angola, the median prediction for 2030 of the low-end range is 22.49 °C (IQR = 21.71, 23.14) and the high-end range is 22.75 °C (IQR = 22.04, 23.43). For 2070, the low-end range is 22.51 °C (IQR = 21.7, 23.29) and the high-end range is 24.89 °C (IQR = 23.6, 26.42). Whereas for 2100, the low-end range is 22.5 °C (IQR = 21.51, 23.16) and the high-end range is 26.92 °C (IQR = 25.41, 29.64) (Figure 8).

For Cameroon, the low-end range median prediction for 2030 is 25.45 °C (IQR = 24.74, 25.89) and the high-end range is 25.61 °C (IQR = 24.98, 26.23). In 2070, the low-end range is 25.6 °C (24.78, 26.25) and the high-end range is 27.6 °C (IQR = 26.14, 28.91). For 2100, the low-end range median projection is 25.56 °C (IQR = 24.68, 26.08), and the high-end range is 29.33 °C (IQR = 27.82, 31.66) (Figure 8).

Central African Republic, the median prediction for the 2030 low-end range is 25.89 °C (IQR = 24.95, 26.43) and the high-end range is 26.05 °C (IQR = 25.2, 26.71). The estimates for the 2070 low-end range are 26.1 °C (IQR = 25.15, 26.91), whereas the high-end range is 27.96 °C (IQR = 25.92, 29.54). The low-end range prediction for 2100 is 26.07 °C (IQR = 25.15, 26.72), and the high-end range prediction is 29.72 °C (IQR = 27.92, 32.61) (Figure 8).

Chad’s low-end range prediction for 2030 is 28.07 °C (IQR = 26.92, 28.75), whereas the high-end range is 28.37 °C (IQR = 27.47, 29.11). For the year 2070, the low-end range is 28.27 °C (IQR = 26.89, 29.15) and the high-end range is 30.59 °C (IQR = 29.05, 32.11). In 2100, the low-end range prediction is 28.06 °C (IQR = 26.87, 28.76) whereas the high-end range is 32.9 °C (IQR = 30.94, 35.12) (Figure 8).

For 2030, Congo’s low-end range is 25.45 °C (IQR = 24.8, 25.97), and the high-end range prediction is 25.59 °C (IQR = 25.04, 26.17). For 2070, the low-end range prediction is 25.53 °C (IQR = 24.85, 26.21) and the high-end range is 27.44 °C (IQR = 26.28, 28.96). In 2100, the low-end range prediction is 25.49 °C (IQR = 24.68, 26), while the high-end range prediction is 29.19 °C (IQR = 27.82, 31.65) (Figure 8).

The Democratic Republic of the Congo’s low-end range prediction for 2030 is 24.99 °C (IQR = 24.26, 25.53), and the high-end range prediction is 25.19 °C (IQR = 24.49, 25.8). In 2070, the low-end range prediction is 25.07 °C (IQR = 24.36, 25.85) and the high-end range prediction is 27.1 °C (IQR = 25.41, 28.74). For 2100, the low-end range prediction is 25.01 °C (IQR = 24.26, 25.6), while the high-end range prediction is 28.95 °C (IQR = 27.21, 31.66) (Figure 8).

Equatorial Guinea’s 2030 low-end range prediction is 25.47 °C (IQR = 24.89, 25.8), whereas the high-end range prediction is 25.6 °C (IQR = 25.17, 26.11). For 2070, the low-end range prediction is 25.52 °C (IQR = 24.92, 26.05), and the high-end range prediction is 27.39 °C (IQR = 26.51, 28.41). In the year 2100, the low-end range prediction is 25.46 °C (IQR = 24.77, 25.95) and the high-end range prediction is 28.97 °C (IQR = 27.68, 30.6) (Figure 8).

In Gabon, the 2030 low-end range prediction is 25.88 °C (IQR = 25.33, 26.24) and the high-end range prediction is 26.03 °C (IQR = 25.6, 26.55). For 2070, the low-end range prediction is 25.94 °C (IQR = 25.33, 26.46), while the high-end range prediction is 27.86 °C (IQR = 26.94, 19.02). For the year 2100, the low-end range prediction is 25.88 °C (IQR = 25.2, 26.35), and the high-end range prediction is 29.51 °C (IQR = 28.18, 31.48) (Figure 8).

Sao Tome and Principe’s 2030 low-end range prediction is 25.86 °C (IQR = 25.43, 26.17), and the high-end range prediction is 25.99 °C (IQR = 25.59, 26.49). The low-end range prediction for 2070 is 25.94 °C (IQR = 25.33, 26.43), and the high-end range prediction is 27.62 °C (IQR = 26.75, 28.57). In 2100, the low-end range prediction is 25.84 °C (IQR = 25.31, 26.2), whereas the high-end range prediction is 29.23 °C (IQR = 27.86, 30.53) (Figure 8).

#### 3.4.2. Statistical Reporting of SPEI Drought Index Projections between 2030 and 2100

For Angola, 2030 predictions suggested a low-end median SPEI index of −0.11 (IQR = 1.26, 0.44), with a high-end median index of 0.02 (−1.1, 0.91). For 2070, the low-end median was −0.18 (−1.56, 0.38) and the high-end median was −0.3 (IQR = −1.84, −0.47). In 2100, the low-end median index was −0.12 (IQR = −1.07, 0.39) and the high-end index was −0.43 (IQR = −2.17, 0.41). All values were near normal; however, the high-end index for 2100 was nearing moderately dry conditions (Figure 9).

In Cameroon, 2030 predictions had a low-end median SPEI index of 0.01 (IQR = −1.25, 0.46), with high-end predictions of 0.04 (IQR = −0.87, 1.29). In 2070, the prediction of the low-end median index was 0.01 (IQR = −1.25, 0.47), and the high-end prediction was 0.09 (IQR = −0.93, 1.57). In 2100, the low-end median index prediction was 0.06 (IQR = −1, 0.47) and the high-end median was 0.01 (IQR = −1.53, 1.7). All median values were near normal; however, the first interquartile range for 2030 suggested moderately dry conditions, and the first interquartile range for 2100 suggested severely dry conditions (Figure 9).

The Central African Republic’s 2030 prediction for the low-end median SPEI index was 0.06 (IQR = −0.88, 0.56), and the high-end index was 0.11 (IQR = −0.66, 1.14). For 2070, the low-end median SPEI index was −0.07 (IQR = −1.16, 0.31) with a high-end median of 0.26 (IQR = −0.53, 1.85). For the year 2100, the low-end median index was 0.02 (IQR = −0.97, 0.55), while the high-end median index was 0.02 (IQR = −1.4, 1.95). Overall, the values were within the near-normal range; however, the interquartile ranges for 2100 reached moderately dry conditions, with periods of moderately wet conditions too (Figure 9).

In Chad, the low-end median SPEI index prediction for 2030 was 0.12 (IQR = −0.88, 0.58), and the high-end index was −0.06 (IQR = −1.15, 0.68). In 2070, the low-end median projection was −0.2 (IQR = −1.66, 0.26) and the high-end index was −0.23 (IQR = −1.48, 0.73). For 2100, the low-end median index projection was −0.02 (IQR = −1.21, 0.42) and the high-end projection was −0.27 (IQR = −2.09, 0.83). Overall, the values were near normal, but the interquartile ranges suggested severely dry periods as well (Figure 9).

For Congo, the low-end median SPEI index prediction for 2030 was −0.11 (IQR = −1.42, 0.26), with the high-end index at 0.06 (IQR = −0.74, 1.17). For 2070, the low-end median projection was −0.06 (IQR = −1.51, 0.52), with the high-end median at 0.04 (IQR = −1.01, 1.57). In 2100, the low-end median projection for SPEI was 0.02 (IQR = −1.16, 0.62) and the high-end projection was 0.01 (IQR = −1.53, 1.87) (Figure 9).

In the Democratic Republic of the Congo, the SPEI index projections for 2030 were −0.03 (IQR = −1.24, 0.47) for the low end, and the high end projections were 0.1 (IQR = −0.7, 1.25). In 2070, the low-end median projection was −0.15 (IQR = −1.43- > 0.29) with the high-end at 0.08 (IQR = −1.02, 1.62). For 2100, the low-end median projection was 0.02 (IQR = −0.83, 0.48), and the high-end projection was 0.02 (IQR = −1.54, 1.72) (Figure 9).

For Equatorial Guinea, the 2030 low-end and high-end SPEI index projections were 0.03 (IQR = −1.43, 0.47), and 0.03 (IQR = −0.83, 1.27), respectively. For 2070, the low-end and high-end median projections were 0.13 (IQR = −0.73, 0.68), and 0.08 (IQR = −1.19, 1.76), respectively. In 2100, the low-end and high-end median projections were 0.13 (IQR = −0.36, 0.85), and 0.09 (IQR = −1.76, 2.19), respectively (Figure 9).

In Gabon, the median low-end SPEI index projection was −0.07 (IQR = −1.21, 0.38) and the high-end projection was 0.06 (IQR = −0.8, 1.21). For 2070, the low-end median projection was 0.06 (IQR = −1.07, 0.66), and the high-end median projection was 0.09 (IQR = −0.94, 1.74). In 2100, the low-end median index projection was 0.06 (IQR = −0.8, 0.86) and the high-end projection was −1.32 (IQR = −1.32, 2.09) (Figure 9).

In Sao Tome and Principe, the 2030 low-end SPEI index projection was −0.1 (IQR = −1.05, 0.22) and the high-end projection was 0.16 (IQR = −0.67, 1.35). For 2070, the low-end projection was 0.02 (IQR = −1.09, 0.92), and the high-end projection was 0.01 (IQR = −0.82, 1.7). For 2100, the median low-end index projection was 0.04 (IQR = −0.84, 1.06) while the high-end projection was 0.07 (IQR = −1.03, 1.65) (Figure 9).

## 4. Discussion

In this population-based original study, findings of food insecurity (acute food insecurity), malnutrition (undernourishment, wasting, stunting, and mortality), and infectious disease (diarrhea and pneumonia-related deaths) were reported between the years 1990 and 2022. Projections for mean air temperature values and the SPEI drought index were made from 2030 through 2100. Overall, increased food insecurity was witnessed in the region, particularly after the onset of the COVID-19 pandemic. Malnutrition indicators suggested a general drop in wasting and stunting among children since 1990; however, the trends were still alarmingly high as compared with the global estimates. Whereas, a comparable trend was witnessed for an adjusted rate of change for mortality due to diarrhea and pneumonia, where a net negative change was present but was still lagging behind the world average. The mean air temperature rises were close to 1–3 °C, and the drought index suggested an overall median close to near normal; however, moderately dry and severely dry estimates were seen in the 10th–90th percentile findings. The SSP1–1.9 and SSP5–8.5 findings were utilized to assess the low-end and high-end ranges, which account for socioeconomic pathways and hence are helpful in accurately predicting the predicted changes in outcomes.

In our study, food insecurity in the Central African Republic, the Democratic Republic of the Congo, Cameroon, and Chad was mapped. While over 2 million were in TIP Phase 3 of the crisis in the Central African Republic, around 0.64 million are in IPC Phase 4 of emergency; this means that the country is in a high state of acute food insecurity, with food displacements in peri-urban and rural households. Whereas, as of May 2022, there are 22.6 million residents in the Democratic Republic of the Congo that are in IPC Phase 3 of crisis, and around 4 million are in IPC Phase 4 of food insecurity; these values represent 21% of the entire population in the Democratic Republic of the Congo. Whereas, in Cameroon, as of May 2022, 9% (*n* = 2,413,288) of the population is in an IPC Phase 3 crisis. Chad’s situation is also comparable with that of 14% (*n* = 2,098,861) of the population experiencing an IPC Phase 3 acute food crisis.

In general, malnutrition is a major cause of death in the pediatric population [46,47]. In total, 6 million children in West and Central Africa are currently affected by severe acute malnutrition (SAM) [48]. Various factors contributed to emergency trends of malnutrition in the region, comprising periodic droughts and climate-change-related shocks, crop and land degradation, low access to essential services and basic food staples, and population growth [49,50,51,52,53,54,55]. Malnutrition is not unidirectional and consists of causes such as infectious diseases such as water-borne pathogenic disease and malaria, limited access to water and sanitation, unhygienic measures, and inadequate child feeding practices [56,57,58,59]. Notable, ready-to-use therapeutic food (RUTF) has brought excellent results in treating malnourished children [60,61,62,63]. The UNICEF estimates that there is scope for a recovery rate of 85–90% if RUTF is widely used in the region; however, issues such as limited access to SAM treatment and late arrival at treatment centers are key barriers to malnutrition treatment [48].

GHI findings raise awareness and understanding of the struggles against hunger and aid in calling attention to the regions with the highest hunger levels. In our study, we found that the Central African Republic and Chad had alarming levels of hunger and ranked 120th and 117^th^, respectively, out of 121 countries. Whereas, Angola and the Congo had serious hunger levels, ranking 98th and 105th out of 121 countries, respectively. Cameroon and Gabon had moderate levels of hunger and ranked 80th and 76^th^, respectively, out of 121 countries. A rapidly changing climate characterized by erratic rains in Central Africa led to a serious drought in 2021, which compounded the humanitarian situation caused by COVID-19, internal conflicts, and plague locusts in the region [64,65].

Central Africa is undergoing rapid growth in economic activities, urbanization, and connectivity [66,67,68]. However, the recent Ebola virus epidemic depicts the vulnerability of the region to newly emerging infectious diseases [69,70]. Prior to 2013, Ebola virus outbreaks in the region consisted of fewer case counts, with predominant rural afflictions [71,72]. However, even if ongoing outbreaks in countries including the Democratic Republic of the Congo are contained, past demographic data indicate that the advancement of human life across Central Africa increases the risk of outbreaks targeting larger populations [71,72]. In recent decades, the Central African Region has undergone population-level changes where the death rate has declined, particularly among children under five [73]. However, one essential factor is the rapid geographic spread of infectious disease due to road construction that opens access to remote locations across Central Africa, among communities that were previously isolated [74]. More recently, the monkeypox virus has increased in frequency in Central Africa, with isolates detected in the Central African Republic and the Democratic Republic of the Congo [75]. The monkeypox virus belongs to the Poxviridae family and is similar to smallpox in clinical presentation [76,77]. With a mortality rate of 1–10%, the monkeypox virus has been isolated twice from wild animals [78,79]. The virus was a growing concern in 2018, when growth was seen in Central Africa and human cases were also reported in western Cameroon [74]. Moreover, the burden of the human immunodeficiency virus (HIV) is also a cause of concern for the overall management of healthcare diseases in the Central African Region [80,81,82]. The World Health Organization’s malaria report for 2020 also stated that around 94% of the world’s malaria cases originate from Africa and bear 95% of the global mortality [83,84,85].

An East African study addressed projected climate change impacts on drought patterns and found that an increase in RCPs over East Africa was anticipated [35]. Moreover, the projected drought was predicted to be greater in May and April of every year through 2100 [35]. However, the study also reported that with uncertainty about climate change impacts on drought patterns, the real-world outcomes may be further worsened by the end of the 21st century [35]. Moreover, as with our study, both wet and dry conditions were noted for Central African countries; what this means is that the “drying will get drier and the wetting will get wetter” [35]. It is imperative to understand drought and mean air temperature projections for the future as they are key resources for infrastructure, policymaking, and climate change countermeasures [86,87,88]. The projected drought conditions predicted in our study suggest that the dry days may be prone to becoming moderately and/or severely dry, whereas the wet days may also become moderately and/or severely wet. The results are intended to provide supporting evidence to local, national, and regional players to plan for mitigating efforts in Central Africa.

The self-sufficiency ratio (SSR) of the African region for food security has reached 0.8 from 1.0 in the past 50 years and has been more volatile than stable [89]. The decrease in sustainability of food may be attributed to the growing gap in, first, food consumption and, second, food production’s growth rate. The region relies on net food imports to reduce the gap in SSR. While Northern and Southern African regions have faced some of the most severe SSR decreases, Central African countries have been slightly more stable than their counterparts [90]. As has been indicated in recent studies, while food production is one of the largest contributors to self-sufficiency in Central Africa, the increased food supply in recent years has helped in limiting large gaps in food supplies. Overall, as the population of Central Africa rises, the income increases, thereby affecting GDP per capita (i.e., an indicator of economic development) [13,14,51]. While the consumption level of Central Africa is much lower than the global average, the augmented demand for food poses problems for the population and places pressure on food security in the future [91]. With more pressure on food production and large impacts of climate change on arable land requirements, there are negative underpinnings on temperature, water, pests, and soil. Overall, there is tension between food production, food demand, and thereby fluctuating food security in Central Africa that is imminent in the future.

### 4.1. Limitations

There are certain limitations to this research article. It is imperative to note that, given the paucity of data reported by regional, national, and international organizations, this is a key limitation of our paper. Similarly, data was not reported for food insecurity in São Tomé and Principe. The DRC, Equatorial Guinea, and São Tomé and Principe did not report on undernourishment or GHI data. These limitations are a key call to action for global health agencies and countries to promote data maintenance and reporting to better estimate the real impacts climate change has on the health crisis. The databases/datasets reported data at differing time periods, which was an unideal scientific synthesis. However, we believe the findings are still pertinent as they depict a current picture of the crises ongoing in the Central African region. Overall, there has been a dearth in both the reporting of data by stakeholders and obtaining pivotal information due to the limited reporting in the region.

### 4.2. Recommendations

For food insecurity, there are central concerns for infrastructure investments, humanitarian aid, livelihood support, and accounting for inter- and intra-community conflicts. The livelihoods of houses in crisis (in IPC Phase 3) or in an emergency (in IPC Phase 4) can be strengthened with access to agricultural avenues and honing technical capacities. To promote infrastructure, food reserves may be promoted with agricultural feeder roads; a focus on reduced transport costs will improve access to food in the Central African Region. Humanitarian aid is required for highly food-insecure and displaced populations at IPC Phase 3 or higher. This is an imperative action plan because it will help in scaling up social safety nets and developing social protection programs. Lastly, conflict within and across countries requires stakeholders to take contingency measures at all relevant levels to protect civilians.

Dominica, a country in the Caribbean, is an acceptable example of what countries must do to address future climate change impacts. In 2017, the country established the Climate Resilience Execution Agency for Dominica (CREAD) database that formed clear objectives and measurable targets across all sectors of the economy [92]. Sudden events such as flooding or cyclones, in addition to long-term air temperature rises, are multiplier threats to the existing vulnerability of residents of the Central African Region—these occurrences exacerbate poverty and fragility in the countries and reverse decades of development. The CREAD database has six key principles that may be adopted in the Central African Region [92]. These include (i) forming resilient foundations through inclusive and rapid development, (ii) enabling adaptation of the residents, (iii) protecting critical public services and assets and adapting land use, particularly the agricultural sector, (iv) increasing community capacity for shock recovery, (v) estimating fiscal and macroeconomic risks, and (vi) allowing efficacious implementation of climate change adaptation actions through continuous monitoring and robust governance structures.

To counteract climate change impacts in the Central African Region, weather-driven mitigation ought to be incorporated into the planning and construction of urban areas. Furthermore, sustainability ought to be a key theme in rural communities. Climate-friendly agricultural sources, including clean, low-carbon energy and micro-irrigation practices, must be subsidized. Accessibility to climate and weather information, particularly for women, who comprise a large portion of the agricultural workforce, must be enforced. Local, Central African-led research must be conducted to understand the grassroots problems. Lastly, intra-African cooperation may be key to relieving disasters and managing internal conflicts.

## 5. Conclusions

In this population-based original study, the Central African Region’s acute food insecurity, malnutrition in the pediatric population, and infectious disease trends, including the adjusted rate of mortality rate change due to diarrhea and pneumonia-induced mortality, were assessed. This was further complemented by mean air temperature and SPEI drought index outcomes throughout the 21st century. Of the six countries analyzed for the global hunger index, there was a moderate to alarmingly high level of hunger as of 2022, with countries ranking between 76th and 120th out of 121 countries. The projected drought and air temperature outcomes are suggestive of overall rises of 1–3 °C by the year 2100. However, with moderately or severely dry periods anticipated largely between April and May of every year, there are also largely wet days anticipated with temperature extremes. With ongoing and expected threats to agricultural practices and human health as a result of climate change, this study provides essential evidence for stakeholders to plan for and implement mitigating efforts in Central Africa. The next directions in this area must be targeted at holistically reducing the direct and indirect impacts of the climate and health crises. Certainly, an improvement in the assessment of disease via public and private health surveillance can reduce the reappearance of infectious diseases.

## Figures and Tables

**Figure 1 ijerph-20-02697-f001:**
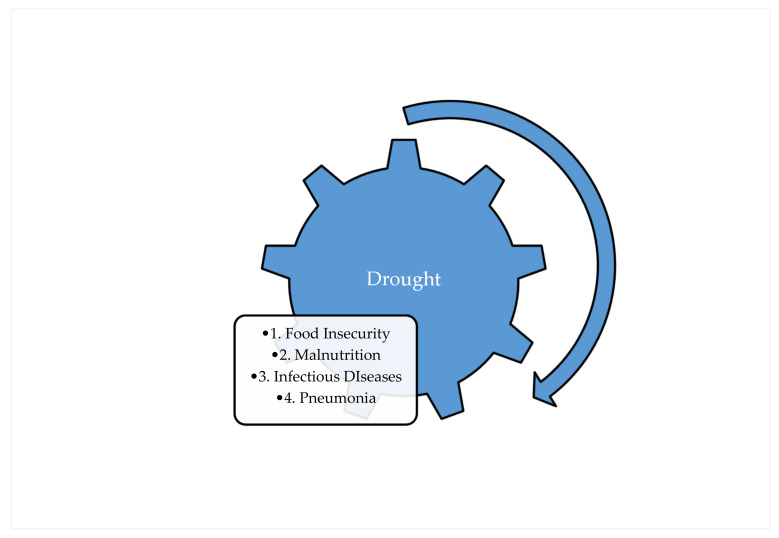
The ‘Drought’ framework proposed by the World Health Organization with four key covariates.

**Figure 2 ijerph-20-02697-f002:**
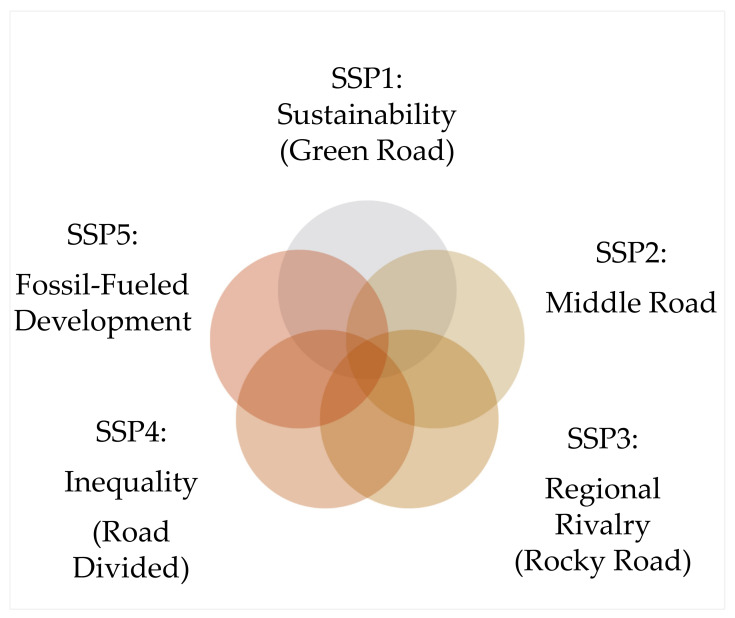
SSP1–5 scenarios.

**Figure 3 ijerph-20-02697-f003:**
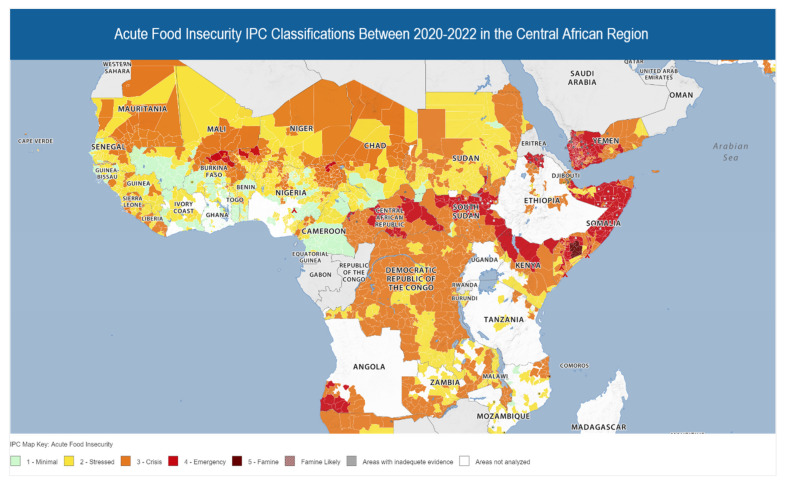
Acute food insecurity classifications based on the IPC between the years 2020 and 2022 in the Central African Region. The image is adapted from the IPC [39].

**Figure 4 ijerph-20-02697-f004:**
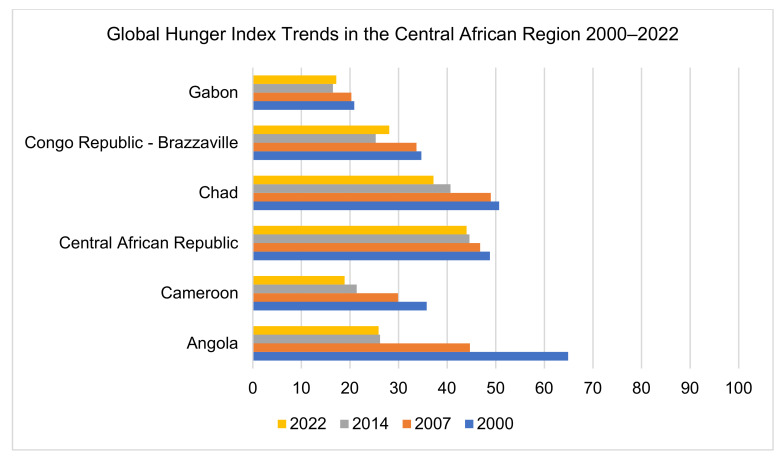
Global hunger index (GHI) trends between 2000 and 2022 across six countries in the Central African Region.

**Figure 5 ijerph-20-02697-f005:**
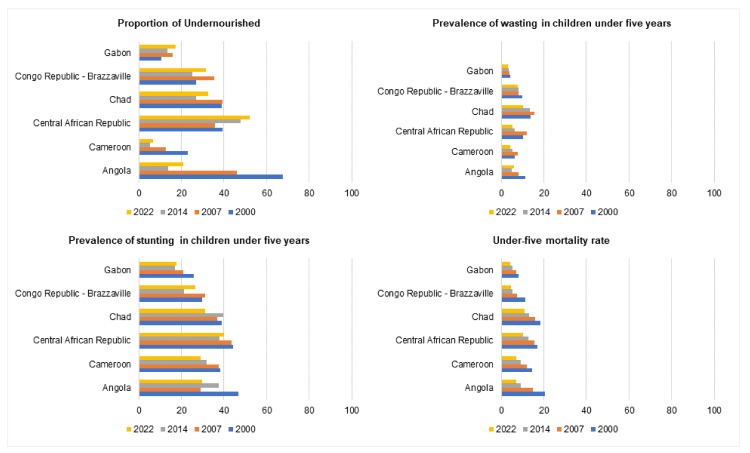
Undernourishment, wasting, stunting, and mortality trends between 2000 and 2022 across six countries in Central Africa.

**Figure 6 ijerph-20-02697-f006:**
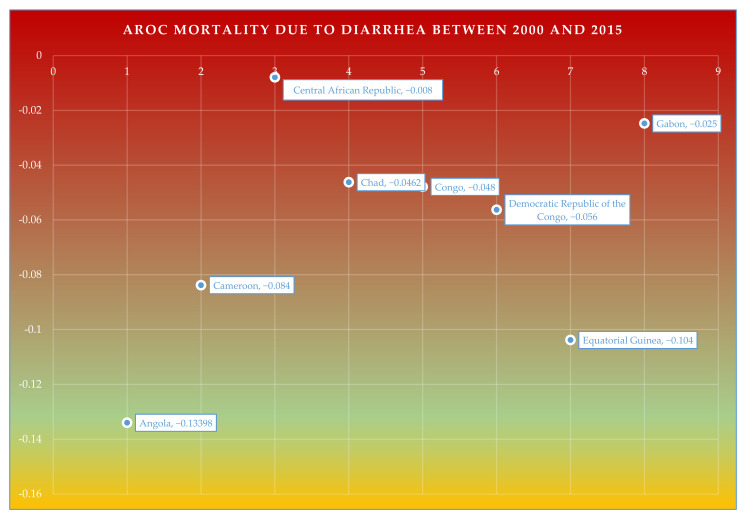
Adjusted Rate of Change (AROC) for mortality due to diarrhea between the years 2000 and 2015.

**Figure 7 ijerph-20-02697-f007:**
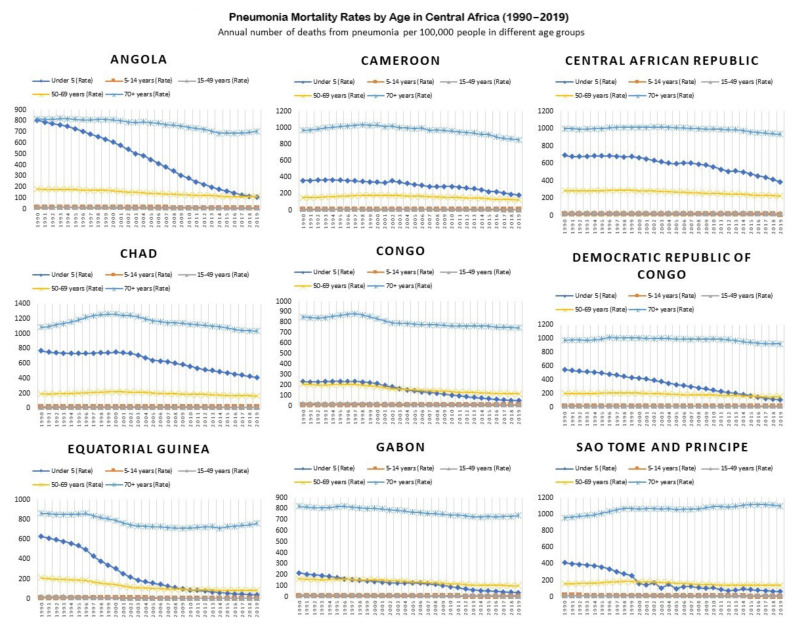
Pneumonia mortality rates in Central Africa between 1990 and 2019 are presented as the annual number of deaths from pneumonia per 100,000 people in different age groups.

**Figure 8 ijerph-20-02697-f008:**
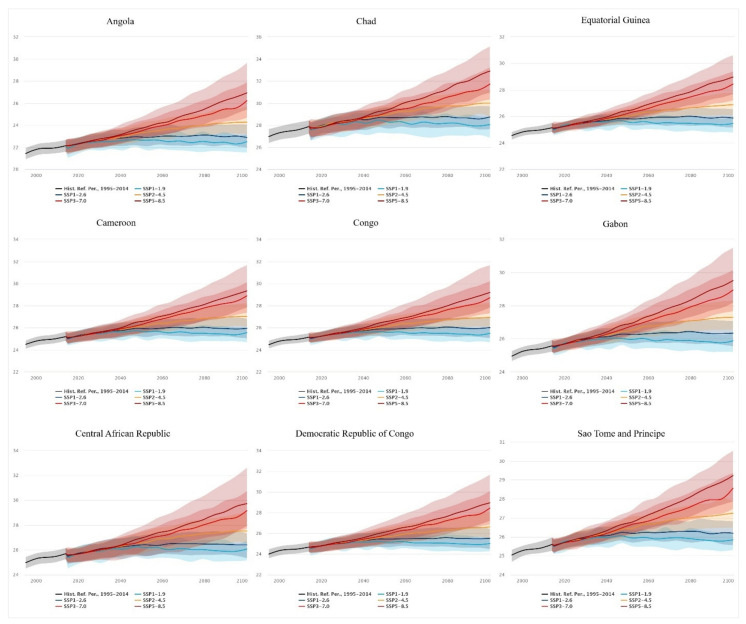
Projected mean air temperature in the Central African Region using a multi-model ensemble with historical reference dated between 1995 and 2014 [43].

**Figure 9 ijerph-20-02697-f009:**
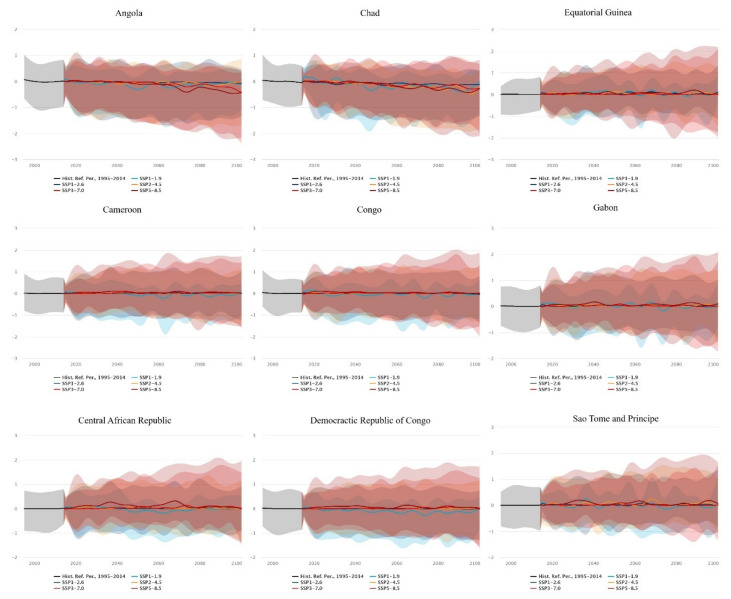
Projected annual SPEI drought index in the Central African Region using a multi-model ensemble with historical reference dated between 1995 and 2014 [43].

**Table 1 ijerph-20-02697-t001:** Database/dataset pools and identifiers.

No.	Database/Dataset Pool	Identifier
1	IPC country analysis	https://www.ipcinfo.org/ipc-country-analysis/en/ (accessed on 3 November 2022)
2	Concern Worldwide and Welthungerhilfe	https://www.globalhungerindex.org/trends.html (accessed on 3 November 2022)
3	Institute for Health Metrics and Evaluation, Global Burden of Disease	Institute for Health Metrics and Evaluation (IHME): Africa Under-5 Diarrhea Incidence, Prevalence, and Mortality Geospatial Estimates 2000–2015
4	Global Burden of Disease Collaborative Network	Roth, G.A. Global Burden of Disease Collaborative Network. Global Burden of Disease Study 2017 (GBD 2017) Results
5	Survey data	Censuses, household, and country-specific survey series (Multiple Cluster Survey, MICS; Demographic and Health Survey, DHS)
6	The World Bank	https://climateknowledgeportal.worldbank.org/ (accessed on 3 November 2022)

**Table 2 ijerph-20-02697-t002:** SSPs in the IPCC AR6 [44].

SSP	Scenario	Estimated Warming (2041–2060)	Estimated Warming (2081–2100)	Very Likely Range in °C (2081–2100)
**SSP1** **–1.9**	Very low GHG emissions: CO_2_ emissions cut to net zero around 2050	1.6 °C	1.4 °C	1.0–1.8
**SSP1** **–2.6**	Low GHG emissions: CO_2_ emissions cut to net zero around 2075	1.7 °C	1.8 °C	1.3–2.4
**SSP2** **–4.5**	Intermediate GHG emissions: CO_2_ emissions around current levels until 2050, then falling but not reaching net zero by 2100	2.0 °C	2.7 °C	2.1–3.5
**SSP3** **–7.0**	High GHG emissions: CO_2_ emissions double by 2100	2.1 °C	3.6 °C	2.8–4.6
**SSP5** **–8.5**	Very high GHG emissions: CO_2_ emissions triple by 2075	2.4 °C	4.4 °C	3.3–5.7

SSP: socioeconomic pathways.

**Table 3 ijerph-20-02697-t003:** The Standardized Precipitation Evapotranspiration Index (SPEI).

Input Variables	Monthly Air Temperature and Precipitation Data Was Used
Application	SPEI was primarily used to both monitor and identify conditions linked with drought impacts
Key Resources	SPEI code (http://spei.csic.es/, accessed on 3 November 2022) by the Consejo Superior de Investigaciones Científicas (CSIC) *; Flood and Drought Portal by GEF, UN Environment, IWA, and DHI (http://www.flooddroughtmonitor.com/ib, accessed on 3 November 2022)

* The SPEIbase is based on monthly precipitation and potential evapotranspiration from the Climatic Research Unit of the University of East Anglia. Currently, version 4.05 of the CRU TS dataset is being used. The SPEIbase is based on the FAO-56 Penman–Monteith estimation of potential evapotranspiration. This is a major difference with respect to the SPEI Global Drought Monitor, which uses the Thornthwaite PET estimation. The Penman–Montheith method is considered a superior method, so the SPEIbase is recommended for most uses, including long-term climatological analysis.

## Data Availability

All data utilized for the purpose of this study are available publicly and online in the enlisted databases. Curated datasets may be requested by the corresponding author (Z.S.) on reasonable request.

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
