# Peer review of "Mean Temperature and Drought Projections in Central Africa: A Population-Based Study of Food Insecurity, Childhood Malnutrition and Mortality, and Infectious Disease"

_ijerph, 2023, doi:10.3390/ijerph20032697_

Round 1

Reviewer 1 Report

Please find my review report attached.

Author Response

To the Reviewer: Before you go on to read the scientific content of this revision, I would humbly like to thank you for volunteering to review this essential paper. Your time, dedication, and efforts are truly and deeply valued.

-Dr. Zouina Sarfraz

Reviewer 1 Comments and Author Responses:

Comments: This study focuses on four key themes including acute food insecurity (AFI), childhood malnutrition and mortality, infectious disease burden, and projections for drought and mean temperature throughout the twenty-first century in nine Central African countries. The goal of this study is to connect these four themes and provide a past, present, and future population-based view of climate impacts, health, and food trends in this geography. The authors look at the historical data for food insecurity, childhood malnutrition/mortality and infectious disease for the selected countries. Later, the authors forecast future temperature (in degree Celsius) and drought (in the standardized precipitation evapotranspiration - SPEI index) using five socioeconomic pathways (SSPs) through the year of 2100. The paper discusses the results and concludes with some policy recommendations and a summary. Overall, this paper needs major improvements. This manuscript includes many information, and it is hard to bring them together. The authors should improve the paper to gather them together in a more coherent way. On the other hand, this manuscript can only focus on one of the important key themes, and develop the analysis and the discussion around it. Some of the important comments are listed below. 

Author Response: I thank you for reading through the paper, providing your expert advice and allowing us to review the paper based on your comments. I have enlisted responses to your comments (point-by-point) below.

Comment 1: Abstract. Some information in the abstract can be removed and I suggest to reorganize the abstract as: The main concept for the study, data/methodology, main results and takeaway for the readers. 

Author Response: Thank you for your constructive feedback. MDPI policies recommend not to include any suborganization of the abstract, which means it should be read as a paragraph of 200 words or less. For this reason, I have removed unnecessary information and the word count now stands at 197. The main results are too large to enter but the last two sentences of the abstract present them. I invite you to review my changes.

Comment 2: Introduction. Explain population-based study. 

Author Response: A definition of population-based studies has been added. In general, these are well understood by readers. It is as follows: “Population-based studies are defined as a cohort of individuals from the general pop-ulation who share common characteristics, such as age, sex, or health conditions.

Comment 3: Introduction. A paragraph should be added to summarize the structure of the paper. 

Author Response: It has been updated along with a framework (WHO) that has originally led us to structure the paper as such. Happy reading.

Comment 4: Materials and Methodology. I could not find any details about the data used for the projections or forecasting methodology in this section. 

Author Response: Thank you for your insightful comment. Please refer to the database/dataset and identifier information in Table 1. Additionally, please see newly added supporting information for projections of forecasting: “SSPs are a component of the new framework adapted in climate change research to fa-cilitate integrated analysis of future climate impacts, adaptation, vulnerabilities and mitigation. The database accounts for the following: i) population by sex, age, and ed-ucation, ii) urbanization, and iii) economic development. The forecasting until the year 2100 is not only limited to basic SSP socio-economic elements; but also includes scenarios by integrated assessment models (IAMs). These scenarios also account for i) land-use, ii) energy supply and use, iii) greenhouse gas emissions and air pollutant emissions, iv) average temperature change, and v) mitigation costs. The climate pro-jection is based on modeled data based on the global climate compilations of the coupled model intercomparison projects (CMIPs). The projection data is presented at a 1.0º x 1.0º (100km x 100km) resolution.

Comment 5: Materials and Methodology. It is not clear how the United Nations Sustainable Development Goals (SDGs) are used for the rest of the analysis. 

Author Response: I have made a clarifying statement. Please see that the UNSDGs focus on 4 indicators to measure hunger. If you look at the 4 points below, they are as follows:

These 4 indicators, as enlisted below, are used to measure the progress of hunger miti-gation efforts toward the United Nations Sustainable Development Goals (SDGs). These comprise of the following (as I have already enlisted in the manuscript):

  1. Undernourishment
  2. Child stunting

iii. Child wasting

  1. Child mortality

Comment 6: Materials and Methodology. SSPs determination and SPEI index calculations are not clearly stated in this section. 

Author Response: Thank you for your very insightful and required comment. If you have a look at the methods section, all your concerns have been addressed at the very core.

Comment 7: Results. Subsections from 3.1 to 3.3 are data summary instead of any sociopolitical analysis. These parts need to be revisited. 

Author Response: Thank you for your comment. However, the only agenda of the results of our study is to present trends and statistics. For this reason, all authors have opted to retain the subsections. The analytical section is within the discussion. It would not be feasible to make sociopolitical commentary of a study of this nature. Thank you.

Comment 8: Figure 2 and 3. The quality of the graph needs to be improved. 

Author Response: Thank you for your comment. The Figures are 600 dpi now.

Comment 9: Some of the data are missing certain countries. The missing data issue should be addressed well and discussed how it could affect the analysis. Some examples: São Tomé & Principe is not included Food Insecurity portion or in the Figure 1; GHI and Undernourishment data do not include DRC as well as Equatorial Guinea and São Tomé & Principe; Only Congo Republic’s capital (Brazzaville) included in GHI and Undernourishment data; São Tomé & Principe is not included in AROC of mortality. 

Author Response: 

You have pointed out a key issue and I fully agree with it. I have added a new limitations section. Please review!

“4.1. Limitations

There are certain limitations of this study. It is imperative to note that given the paucity of data reported by regional, national, and international organizations, it is a key limitation to our paper. Similarly, data was not reported for food insecurity in São Tomé & Principe. The DRC, Equatorial Guinea and São Tomé & Principe did not report on undernourishment and GHI data. These limitations are a key call to action for global health agencies and countries to promote data maintenance and reporting to better estimate the real impacts climate change has on the health crisis. The data-bases/datasets reported data at different time periods which was an unideal scientific synthesis. However, we believe the findings are still pertinent as they depict a current picture of the crises ongoing in the Central African region. Overall, there has been a dearth in both reporting of data by stakeholders and obtaining pivotal information due to the limited data-reporting in the region.”

Comment 10: Figure 4 is hard to follow. A different chart option can be used. 

Author Response: Thank you for your insightful comment. The chart has been changed entirely.

Comment 11: Results. It is not clear why the authors choose to forecast through the year of 2100. Any studies forecasting for a longer period requires a more thorough explanation of methodology and techniques to check the robustness of the analysis.

Author Response: Please refer back to my responses to comments 5 and 6 to the whys. That will answer this entire comment.

Comment 12: Results. IQR’s should be explained in detail at the methodology part. 

Author Response: Thank you very much for your comment. In general, it is assumed that IQRs are a part of the report. However, I have added the following to ensure your comment is addressed: “The values were reported with interquartile ranges (IQR), wherever applicable. The IQR is a measure of statistical dispersion, which was utilized to report the spread of the data findings.”

Comment 13: Discussion. Food trends were not discussed in this section although it was included as a part of the goal for the study.

Author Response: Thank you for giving me solid advice on making this paper the strongest the world will ever see. Here are my additions: “The self-sufficiency ratio (SSR) of the African region for food security reached 0.8 from 1.0 in the past 50 years and has been more volatile than stable. The decrease in sustainability of food may be attributed to the growing gap in, at first, food consump-tion, and at second, food production’s growth rate. The region relies on net food im-ports to reduce the gap in SSR. While Northern and Southern African regions have faced some of the most severe SSR decreases, Central African countries have been slightly more stable than their counterparts. As has been indicated in recent studies, while food production is one of the largest contributors to self-sufficiency in Central Africa, the increased food supply in recent years has helped in limiting large gaps in food supplies. Overall, as the population of Central Africa rises, the income increas-es, thereby affecting GDP per capita (i.e., an indicator of economic development) [13, 14, 51]. While the consumption level of Central Africa is much lower than the global average, the augmented demand for food poses problems for the population and places pressure on food security in the future. With more pressure on food production and large impacts of climate change on arable land requirement, there are negative under-pinnings on temperature, water, pests, and soil. Overall, there is tension between food production, food demand and thereby fluctuating food security in Central Africa that is imminent in the future.”

Reviewer 2 Report

I do congratulate the authors for this exciting paper.

The authors analyze Mean Temperature and Drought Projections in Central Africa, connecting four themes (food insecurity, malnutrition, infectious disease, and climate change), providing a past, present, and future population-based view.

It is mandatory to improve the quality of all Figures, mainly Figures 5, 6, and 7 - it nos possible to read them. In the maps, please include a north arrow and a scale.

The authors should present a summary table in the Materials and Methods section - there is a lot of information, and a summary table would be very welcome.

Congratulations on the section Recommendations - this is really interesting.

The Conclusions section should be improved, resuming the paper's goals and highlighting the paper's limitations and next steps.

Author Response

To the Reviewer: Before you go on to read the scientific content of this revision, I would humbly like to thank you for volunteering to review this essential paper. Your time, dedication, and efforts are truly and deeply valued.

-Dr. Zouina Sarfraz

Reviewer 2 Comments and Author Responses:

I do congratulate the authors for this exciting paper.

The authors analyze Mean Temperature and Drought Projections in Central Africa, connecting four themes (food insecurity, malnutrition, infectious disease, and climate change), providing a past, present, and future population-based view.

Author Response: Thank you for your comments.

Comment 1: It is mandatory to improve the quality of all Figures, mainly Figures 5, 6, and 7 - it is not possible to read them. In the maps, please include a north arrow and a scale.

Author Response: All figures have been improved for quality. Beyond this, the software cannot improve them, which I believe will be further improved during proofing/publication.

Comment 2: The authors should present a summary table in the Materials and Methods section - there is a lot of information, and a summary table would be very welcome.

Author Response: Thank you for your insightful comment. As you can now see, all important information has been tabulated and presented in a palpable manner.

Comment 3: Congratulations on the section Recommendations - this is really interesting.

Author Response: Thank you very much for your comments.

Comment 4: The Conclusions section should be improved, resuming the paper's goals and highlighting the paper's limitations and next steps.

Author Response: It has been updated. I urge and invite you to read the newly added information! Also, a separate limitations section has been retained within the discussion. Do see that too! Thank you.

Reviewer 3 Report

1. The title of the paper, as well as its structure have to be changed. The authors do not analyze the influence of air temperature or drought on food insecurity, childhood malnutrition and mortality, and infectious disease. They just separately describe every of these characteristics. Therefore, it is even not clear why all these topics are combined in the same paper. It would be interesting to make a research of the influence of changes of air temperature or frequency and intensity of droughts on food security, childhood malnutrition and mortality, and infectious disease, but authors do not make such analyses. This paper does not make any scientific sense now. It is just a description of statistical data.

2. In the Introduction, there are too many references on the reports of different organizations instead of references on the scientific papers.

3. It is well-known that there are a lot of types of temperature (air temperature, surface temperature, mean radiant temperature etc.). Authors analyze air temperature, but they did not specify this, however it has to be named correctly in the scientific paper.

4. Authors write in the chapter "Materials and Methods" that "data spanned 1990 until 2022", but acute food insecurity classifications based on the IPC shown for the period 2020-2022, global hunger index (GHI) - for the period 2000-2022, adjusted Rate of Change (AROC) for mortality due to diarrhea - for the period 2000-2015. Authors should write more correctly which data and for which period were used for the paper.

5. The chapter "Discussion" is very poor. The authors mainly describe their own results. The analysis of cause and effect relationships, as well as comparison with the result of the other research are absent. It is not clear where the scientific results in this paper.

6. The authors do not do not explain or justify why they decided to analyze pneumonia and how it is connected with air temperature (and its changes) in Central Africa (as it is a hot region). The authors should add to the "Introduction" an overview about the most dangerous infectious diseases in Central Africa and connections between air temperature, climate and rate of these diseases.

7. In the chapter "Materials and Methods", there is no information about climate projections which were used in the paper. The authors have to mention from which project the data was used, which scenarios, the grid resolution, which were the forecast periods and the basic (historical) period etc.

8. There is no analysis of changes in air temperature in the p.3.4.1 "Temperature Change Predictions Between 2030 and 2100". Just 9 identical paragraphs, which differ only by values of temperature and the title of countries. The absolutely the same with the paragraph "3.4.2. SPEI Drought Index Projections Between 2030 and 2100". It is not suitable for scientific papers. It is a style of statistical data report.

9. The "Conclusions" are not supported by the results.

Author Response

To the Reviewer: Before you go on to read the scientific content of this revision, I would humbly like to thank you for volunteering to review this essential paper. Your time, dedication, and efforts are truly and deeply valued.

-Dr. Zouina Sarfraz

Reviewer 3 Comments and Author Responses:

Comment 1: The title of the paper, as well as its structure have to be changed. The authors do not analyze the influence of air temperature or drought on food insecurity, childhood malnutrition and mortality, and infectious disease. They just separately describe every of these characteristics. Therefore, it is even not clear why all these topics are combined in the same paper. It would be interesting to make a research of the influence of changes of air temperature or frequency and intensity of droughts on food security, childhood malnutrition and mortality, and infectious disease, but authors do not make such analyses. This paper does not make any scientific sense now. It is just a description of statistical data.

Author Response: Respected reviewer, your comment stated “the authors do not analyze the influence of air temperature or drought on food insecurity.” However, I believe you may have misread the title which is as follows: “Mean Temperature and Drought Projections in Central Africa: A Population-Based Study of Food Insecurity, Childhood Malnutrition and Mortality, and Infectious Disease”

Please note that the title does not at all hint that we are interlinking analyses of these different factors. Which is why it is important that you also read the title “POPULATION-BASED STUDY.” I repeat; I did not write analytical for the reason that this is a “SNAPSHOT” study. You are correct; it is ‘just a description of statistical data,’ which is exactly what the study aimed to do. My sincerest advice is that you surely review your comment and my paper again if you wish to gain more scientific clarity on the paper type. One last response to your comment: not once has the word analytical even been used in the manuscript. Please re-read at your convenience.

Comment 2: In the Introduction, there are too many references on the reports of different organizations instead of references on the scientific papers.

Author Response: I do not understand the concern raised by this comment. The topic of climate change is very very scarce particularly in Africa. You should comprehend that as a very important aspect of WHY we are writing a full-fledged scientific paper. The reports that we have referenced were also very difficult to find. In fact, our effort to put together so many important pieces of information from different organizations further strengthens the paper. While I appreciate your comment on the subject matter, I humbly disagree with the rationale behind your comment.

Comment 3: It is well-known that there are a lot of types of temperature (air temperature, surface temperature, mean radiant temperature etc.). Authors analyze air temperature, but they did not specify this, however it has to be named correctly in the scientific paper.

Author Response: Thank you very much for your comment, it is indeed very helpful to point out. The data has been used for “air” temperature. As a result, I have made an update throughout the paper to reflect that we are indeed talking about air temperature. Thank you for your due diligence.

Comment 4: Authors write in the chapter "Materials and Methods" that "data spanned 1990 until 2022", but acute food insecurity classifications based on the IPC shown for the period 2020-2022, global hunger index (GHI) - for the period 2000-2022, adjusted Rate of Change (AROC) for mortality due to diarrhea - for the period 2000-2015. Authors should write more correctly which data and for which period were used for the paper.

Author Response: Thank you very much for your comment. For your comment regarding the date, every figure and finding in the result clearly displays the date. However, for clarity's sake, I have added the following statement in the methods: The cumulative data spanned 1990 until 2022, and projections were made until 2100. However, specific outcome data had different origin- and end-points, which are enlisted in the results section separately for every variable of interest.”

Comment 5: The chapter "Discussion" is very poor. The authors mainly describe their own results. The analysis of cause and effect relationships, as well as comparison with the result of the other research are absent. It is not clear where the scientific results in this paper.

Author Response: Newly supported information has been added. The total discussion now stands at nearly 1600-1800 words. Please do understand that not many studies have been published (in fact, barely 2-3) in this area. We are making the most of the data we found and are ensuring our paper sets us apart with the plethora of sources and data we collected from hundreds of searches! :) 

Comment 6: The authors do not do not explain or justify why they decided to analyze pneumonia and how it is connected with air temperature (and its changes) in Central Africa (as it is a hot region). The authors should add to the "Introduction" an overview about the most dangerous infectious diseases in Central Africa and connections between air temperature, climate and rate of these diseases.

Author Response: For the first part of comment 6-> The reason for analyzing all four co-variates and drought has been made crystal clear in Figure 1. I invite you to review the fact that the World Health Organization released a special report notifying these factors as key in the current ‘climate and health crisis.’ To ensure you review the material, here is a view-online drive PDF document of the report: https://drive.google.com/file/d/1VaX2siEESdk0YE42JBHe0c3TUMfHiLW4/view?usp=sharing (Please scroll to page 3). Thank you~

For the second part of comment 6-> It has been updated. I urge and invite you to read the newly added information!

Comment 7: In the chapter "Materials and Methods", there is no information about climate projections which were used in the paper. The authors have to mention from which project the data was used, which scenarios, the grid resolution, which were the forecast periods and the basic (historical) period etc.

Author Response: I invite you to review the newly added figures and tables and supporting information. I trust this will clarify everything that has been done to conduct this study. Thank you very much for doing your due diligence.

Comment 8: There is no analysis of changes in air temperature in the p.3.4.1 "Temperature Change Predictions Between 2030 and 2100". Just 9 identical paragraphs, which differ only by values of temperature and the title of countries. The absolutely the same with the paragraph "3.4.2. SPEI Drought Index Projections Between 2030 and 2100". It is not suitable for scientific papers. It is a style of statistical data report.

Author Response: Thank you for your comment. The statistics are intentional. The reason they are retained is because such trends will NOT be found in literature. Central Africa (to the best of my knowledge) has no such paper/or even of a similar theme that reports any such data. By the way, this is not meant to be an ‘analysis.’ it is a snapshot.

To ensure I do not mislead the reader, I have this as the subheading 3.4.1. And 3.4.2: “Statistical Reporting of”

The heading in itself has been changed to: 3.4. Statistical Reporting of Air Temperature Changes and Drought Outcomes Through 2100.

Comment 9: The "Conclusions" are not supported by the results.

Author Response: It has been updated. I urge and invite you to read the newly added information!

Round 2

Reviewer 1 Report

The author(s) responded all my comments.

- I still suggest improving the quality of the figures for the publication.

- It would also improve the soundness of the paper if the manuscript explains how author(s) utilize "Data base: SPEI, The Standardised Precipitation-Evapotranspiration Index (csic.es)" specifically for this study. How the author(s) differ using the SPEI from the other researchers using this database. 

Author Response

Comment 1: The author(s) responded all my comments.

Response to Comment 1: Thank you for your due diligence and attention.

Comment 2: I still suggest improving the quality of the figures for the publication.

Response to Comment 2: The maximum resolution of output has been provided.

Comment 3: It would also improve the soundness of the paper if the manuscript explains how author(s) utilize "Data base: SPEI, The Standardised Precipitation-Evapotranspiration Index (csic.es)" specifically for this study. How the author(s) differ using the SPEI from the other researchers using this database.

Response to Comment 3: In response to your very valuable comment, I have added a special note under Table 3 (highlighted in yellow) as follows:

"* The SPEIbase is based on monthly precipitation and potential evapotranspiration from the Climatic Research Unit of the University of East Anglia. Currently the version 4.05 of the CRU TS dataset has been used. The SPEIbase is based on the FAO-56 Penman-Monteith estimation of potential evapotranspiration. This is a major difference with respect to the SPEI Global Drought Monitor, that uses the Thornthwaite PET estimation. The Penman-Montheith method is considered a superior method, so the SPEIbase is recommended for most uses including long-term climatological analysis."

Reviewer 3 Report

-

Author Response

Thank you for your approval of our manuscript. Your volunteering work is greatly appreciated.